# Pleurotus Ostreatus and Volvariella Volvacea Can Enhance the Quality of Purple Field Corn Stover and Modulate Ruminal Fermentation and Feed Utilization in Tropical Beef Cattle

**DOI:** 10.3390/ani9121084

**Published:** 2019-12-04

**Authors:** Benjamad Khonkhaeng, Anusorn Cherdthong

**Affiliations:** Tropical Feed Resource Research and Development Center (TROFREC), Department of Animal Science Faculty of Agriculture, KKU, Khon Kaen 40002, Thailand; k.benjamad211223@gmail.com

**Keywords:** anthocyanin, white-rot fungi, roughage, ruminant, feeding

## Abstract

**Simple Summary:**

The use of purple field corn stover contains enriched anthocyanin. Moreover, the effect of anthocyanin in purple field corn could inhibit methane (CH_4_) synthesis by changing hydrogen from the CH_4_ route to form propionic acid. However, purple field corn was found to have poor digestibility owing to its fiber linkage structure. *Pleurotus ostreatus* and *Volvarialla volvacea* are species of white-rot fungal mushroom, famous and abundant in tropical regions. These fungi have potential to decrease indigestion of cell wall contents and increase cell wall digestibility of rice straw material by secondary metabolites namely lignin peroxidase, manganese peroxidase, and laccase enzyme, and showed the structural polymer found in rice straw and purple field corn residue. Accordingly, feeding purple field corn stover fermented with *P. ostreatus* or *V. volvacea* will increase feed utilization and reduce CH_4_ production in beef cattle.

**Abstract:**

This objective is to elucidate the effect of purple field corn stover treated with *Pleurotus ostreatus* and *Volvarialla volvacea* on feed utilization, ruminal ecology, and CH_4_ synthesis in tropical beef cattle. Four male Thai native beef cattle (100 ± 30 kg of body weight (BW) were assigned randomly as a 2 × 2 factorial arrangement in a 4 × 4 Latin square design. Factor A (roughage sources) was rice straw and purple field corn stover and factor B was species of white-rot fungi (*P. ostreatus* and *V. volvacea*). After fermentation, crude protein (CP) was increased in rice straw and purple field corn stover fermented with *P. ostreatus* and *V. volvacea*. The unfermented purple field corn stover contained 11.8% dry matter (DM) of monomeric anthocyanin (MAC), whereas the MAC concentration decreased when purple field corn stover was fermented with white rot fungi. There were no changes (*p* > 0.05) in DM intake of body weight (BW) kg/d and g/kg BW^0.75^ among the four treatments. The organic matter (OM), CP, and acid detergent fiber (ADF) intake were different between rice straw and purple field corn stover and were the greatest in the purple field corn stover group. Moreover, the current study showed that neutral detergent fiber (NDF) and ADF digestion was higher in purple field corn stover than in rice straw, but there were no significant differences between *P. ostreatus* and *V. volvacea*. There were significant effects of roughage sources on ammonia nitrogen (NH_3_-N) at 4 h after feeding. Bacterial population was changed by feeding with purple field corn stover fermented with *P. ostreatus* and *V. volvacea*. On the other hand, the number of protozoa was reduced by approximately 33% at 4 h after feeding with purple field corn stover (*p* < 0.01). Propionic acid concentration was different between roughage sources (*p* < 0.01) enhanced with purple field corn stover fermented with *P. ostreatus* and *V. volvacea*. In addition, methane production decreased by 15% with purple field corn stover fermented with *P. ostreatus* and *V. volvacea* compared to rice straw. There were significant differences on all nitrogen balances parameters (*p* < 0.05), except the fecal N excretion (*p* > 0.05) were not changed. Furthermore, microbial crude protein and efficiency of microbial N synthesis were enhanced when purple field corn stover fermented with *P. ostreatus* and *V. volvacea* was fed compared to rice straw group. Base on this study, it could be summarized that *P. ostreatus* or *V. volvacea* can enhance the quality of purple field corn stover and modulate rumen fermentation and feed digestion in Thai native beef cattle.

## 1. Introduction

Methane (CH_4_) is a dynamic greenhouse gas synthesized from methanogenic bacteria in the rumen of ruminants. Greenhouse gases are produced through the normal process of feed digestion, and a significant loss of 2 to 12% of feed energy has been reported, which increases feed costs [1]. Scientists have demonstrated a potential approach to reducing CH_4_ emissions with secondary metabolite compounds such as tannins, saponin, or naturally substantive compounds present in some forage crop [1,2].

Purple field corn contains enriched anthocyanin in the cob, husk, silk, and stover [2]. Anthocyanin has a high level of a stronger 2,2-diphenyl-1-picrylhydrazyl scavenging activity [3], making it a beneficial alternative fiber source for ruminant [4]. Moreover, the effect of anthocyanin in purple field corn could inhibit CH_4_ synthesis by changing hydrogen from the CH_4_ route to form propionic acid [5,6]. However, purple field corn was found to have poor digestibility owing to its fiber linkage structure [7]. Thereby, feeding low-quality roughage leads to insufficient nutrients for ruminant to maintain a high production level [8]. Recently, many studies have presented the various biological treatments to improve low-quality roughage as ruminant feed [7,8,9]. White-rot fungi-treated roughage sources are more attractive, particularly in terms of feed utilization, environmentally friendly, exhibit residual substance affect, and are cost effective [10].

*Pleurotus ostreatus* and *Volvarialla volvacea* are species of white-rot fungal mushroom, famous and abundant in tropical regions. They can grow well on rice straw and decrease its structural fiber [10]. These fungi have potential to decrease indigestion of cell wall contents and increase cell wall digestibility of rice straw material by secondary metabolites namely lignin peroxidase, manganese peroxidase, and laccase enzyme, and showed the structural polymer found in rice straw and purple field corn residue [11]. Moreover, lovastatin can be produced by various white-rot fungi by controlling the culture conditions. The lovastatin has been demonstrated to effectively inhibit the HMG-CoA reductase enzyme in methanogenic bacteria by interfering with the assembly of isoprenoid chains needed for membrane phospholipid synthesis and thus selectively depressing the growth of methanogenic bacteria without affecting other ruminal microorganism community [12,13,14]. Our previous work found that purple field corn stover fermented with *P. ostreatus* and *V. volvacea* can be beneficial when fed as an alternative fiber feed to improve in vitro digestibility, rumen fermentation, and reduction of CH_4_ production [12]. However, there is a lack of information on whether purple field corn fermented with *P. ostreatus* and *V. volvacea* is effective in beef cattle. The hypothesis is that feeding cattle with purple field corn stover fermented with *P. ostreatus* or *V. volvacea* will increase feed utilization and reduce CH_4_ production when compared to rice straw. Thus, our objective is to elucidate the effect of purple field corn stover treated with *P. ostreatus* or *V. volvacea* on feed utilization, ruminal ecology, and CH_4_ synthesis in tropical beef cattle.

## 2. Materials and Methods

### 2.1. Animals Care

The study was done at Tropical Feed Resources Research and Development Center, Khon Kaen University. All animals were allowed by the Institutional Animal Care and Use Committee of Khon Kaen University (Animal Ethics No. ACUC-KKU 99/2560), conditional the Animal Experiment Ethics of National Research Council, Thailand.

### 2.2. Animal and Treatments

Four male Thai native beef cattle (100 + 30 kg of body weight (BW) were fed individually in a pen with clean drinking water. The animals were assigned randomly as a 2 × 2 factorial arrangement in a 4 × 4 Latin square design. All animals were fed a concentrate (Table 1) at 1.0% BW. Factor A (roughage sources) was rice straw and purple field corn stover and factor B was species of white-rot fungi (*Pleurotus ostreatus* and *Volvarialla volvacea*).

### 2.3. Preparation of Rice Straw, Purple Field Corn Residue, and Fungal-Treated Substrates

Rice straw was harvested from around rice fields in the area of Khon Kaen, Thailand. Purple field corn residue stover was collected from the Plant Breeding Research Center for Sustainable Agriculture, Faculty of Agriculture, Khon Kaen University. Roughage sources were collected, cut into 2 to 3 cm lengths, and sun dried for 3 to 5 days, then collected in plastic bag and preserved at 28 to 32 °C until used.

*P. ostreatus* and *V. volvacea* were received from the commercial products and were used to inoculate the substrates. They were cultured on potato dextrose agar plate at 25 °C for 7 days, kept at 4 °C, and sub-cultured every couple of weeks. Spore suspension was performed in 0.1% Tween-80 solution at a concentration of nearly 10^7^ spores/mL [7].

The white-rot fungi-treated roughages were handled as follows [14]. First, 200 g of roughage sources were mixed with 200 mL of solution (1% molasses and 1% urea in 200 mL of distilled water) to provide moisture condition of approximately 50% dry matter (DM). Second, roughage, solution and white rot fungi were mixed for around 30 min in TMR machine. After that, the substrates were packed in plastic bags, and kept at room temperature (28 °C to 32 °C) until the mushroom mycelia grew to full colonization and figured on the roughages (about 21 days). Roughages were fed to animals ad libitum. Clean water and mineral block were provided at all times for the whole study. Feeds were fed twice daily at 6:00 am and 4:00 pm.

### 2.4. Feed, Feces, Urine, Rumen Fluid, and Blood Sample Collections

Feed intake was determined daily by weighing the provided and refused feeds before morning feeding. The study was separated into four periods, which each consisted of 21 days. The animals were in an adjusting dietary phase for the first 14 days, and the last 7 days of each phase were for feces and urine sampling as cattle were transferred to metabolic stalls for determination of feed digestion and nitrogen balance by using for total collection method. The fecal samples were sampled about 5% of total fresh weight and divided into two parts; first part for DM analysis every day and second part was kept in refrigerator. Urine was sampled in 10-L vessels containing 10% H_2_SO_4_, to ensure that the pH was decreased below 3.0 in order to prevent the bacterial destruction of purine derivative and creatinine. Urine vessels were replaced every 24 h, the volume of acidified urine was measured and a subsample of 100 mL per animal and was immediately frozen at −20 °C. Feces and urine were pooled by cattle at the end of each period for chemical analysis. During the last 7 days of each period, feed samples (offered and refused) and feces were collected for chemical composition analysis and estimation of feed digestion abilities. The 5% of total fresh weight of fecal matter was collected and separated into two portions: the first portion was analyzed for DM daily, and the latter portion was frozen in a refrigerator and mixed by animals on the last day of each period for nutritional content determination.

The DM of the diets and feces was measured by drying samples at 100 °C for 24 h. The feed and feces samples were incubated by drying in a 60 °C hot air oven. Those samples were ground to 1-mm size. The organic matter (OM) of the samples was analyzed by burning samples at 550 °C. The crude protein (CP) and crude ash of the diet and fecal samples were determined by the protocols of Association of Official Agricultural Chemists (AOAC) [15]. The neutral detergent fiber (NDF) and acid detergent fiber (ADF) were measured by ANKOM^200^ Fiber Analyzer (ANKOM Technology Corporation, Fairport, NY, USA) according to Van Soest et al. [16] methods. The ruminal fluid was sucked by a vacuum pump from the rumen on the last day of each period and approximately 55 mL at 0 and 4 h post morning feeding and divided into three portions. The first part was evaluated for ruminal pH and temperature using a pH meter (Hanna Instruments HI 8424 microcomputer, Singapore). The second portion was examined for rumen ammonia nitrogen (NH_3_-N) concentration using Kjeltech Auto 1030 Analyzer [15] and volatile fatty acids (VFA) using high-performance liquid chromatography (HPLC) [17]. The CH_4_ concentration was run on the equation of Moss et al. [18] based on VFA concentration profiles.

In the final portion, 1 mL ruminal fluid was added to a 9-mL solution of formaldehyde and analyzed for direct counts of total bacteria and the protozoal population using a hemocytometer [19].

Blood samples were taken from the jugular vein (0 and 4 h post feeding on the last day of each period) into tubes with 12 mg of ethylenediaminetetraacetic acid (EDTA). Then, they were put on ice for 30 min, centrifuged at 1500 × *g* at 4 °C for 15 min to separate the plasma, and kept at −20 °C until used to determine blood urea nitrogen (BUN) [20].

Nitrogen balance and purine derivative were measured using the AOAC method [15], although allantoin concentration was evaluated using HPLC. The efficiency of microbial N synthesis (EMNS) concentration was calculated from purine derivative excretions using the equation provided by Chen and Gomes [21]. Microbial crude protein (MCP) was determined using the method proposed by Cherdthong et al. [22].

### 2.5. Measurement of Monomeric Anthocyanin Content (MAC)

The concentration of monomeric anthocyanin (MAC) in the feeds was measured using the method of Lee [23]. A UV–vis spectrophotometer (GENESYS 10S, Thermo Scientific, Waltham, MA, USA) was used to determine MAC at 510 and 700 nm of absorbance in a cuvette with a 1-cm path length. Total MAC per stover DW of one ear (MAC/e) was reported as milligrams of cyanidin-3-glucoside equivalents per 100 g DW (mg CGE/100 g DW) of samples.

### 2.6. Measurement of Lovastatin Content

Lovastatin was analyzed by the method used by Pattanagul [24]. In brief, the lovastatin production is as follows: (i) A total of 1 g sample was extracted with 5 mL of 68% ethanol and shaken at 200 rpm for 1 h, then left to stand in 40 °C for 12 h; (ii) the suspension was centrifuged at 2000 × *g* at room temperature for 15 min and subsequently filtered with Whatman filter No. 1, and the filtrate was dried in a vacuum oven at 55 °C (16 h) and then dissolved with 1 mL of mobile phase solution by filtering through a 0.45 µm nylon membrane into a 2-mL vial to be used for lovastatin measurement by HPLC.

### 2.7. Assay of Statistic

Statistical analyses were performed using a 2 × 2 factorial arrangement in a 4 × 4 Latin square design by using the PROC GLM [25]. The statistical model included roughage source (rice straw and purple field corn stover) fermented with white-rot fungi (*P. ostreatus* and *V. volvacea*) and their interactions. The mean differences between treatments were measured using Duncan’s new multiple range test.

## 3. Results and Discussion

### 3.1. Nutrient Content of Feeds

The feed ingredients and nutrient content of rice straw, purple field corn stover, and concentrate diet are reported in Table 1. The raw rice straw was lower in CP than purple field corn stover, whereas purple field corn stover had high fiber content. After fermentation, CP was increased in rice straw and purple field corn stover fermented with *P. ostreatus* and *V. volvacea*. On the other hand, NDF and ADF in purple field corn stover were significantly higher than in rice straw. It could be due to their stronger fiber linkage structure as lignocellulose. A similar finding was reported by Min [26], who found that the corn stem had the highest lignin content (20.6%) when compared with cob and leaf.

The unfermented purple field corn stover contained 11.8% DM of MAC, whereas the MAC concentration decreased when purple field corn stover was fermented with white rot fungi. This was similar to the finding of a previous study that the MAC in purple field corn stover had a value of 13.49% DM; its variety had a higher genotype of MAC than normal corn varieties [2]. After fermentation, the MAC was decreased, probably because of the high temperature, low pH, and fermentation process [27]. However, the pH condition did not clarify the result of the anthocyanin consistency in the treated substrate, which did not continue to decline. To realize MAC stability during roughage-treated storage, the relationship between MAC stability and lactic fermentation should be explained, because MAC contains anthocyanidin and sugars, and there is a possibility that sugars from MAC are used as a substrate for lactic fermentation [28,29]. The lovastatin concentration was 32.49 to 33.96 g/kg DM when roughages were fermented with *P. ostreatus* and *V. volvacea*, closely studied by Khonkhaeng [12]. Lovastatin can be produced by *P. ostreatus* and *V. volvacea* by controlling the culture conditions. The increase in moisture content also reduced the lovastatin production level. This was probably due to reduced oxygen obtainability caused by redundant substitution of air by water, whereas the low moisture content led to a decreased metabolic heat in the ensiling processing [30]. The pH optimization could support the lovastatin synthesis during ensiling processing. Furthermore, temperature is probably another factor affecting the productivity relating to the stimulation of the enzyme for the synthesis of lovastatin [31]. The temperature at 30 °C revealed that lovastatin could maximize the production; this was noted to be the optimum temperature [32]. Thus, culturing roughages with optimum temperature could result in high concentration of lovastatin [33,34].

### 3.2. Feed Utilization and Nutrient Digestion

The average of roughage, concentrate, and total intake did not change among treatments (*p* > 0.05) (Table 2). There were no changes (*p* > 0.05) in DM intake of BW kg/d and g/kg BW^0.75^ among the four treatments. Even though anthocyanins are phenolic compounds that contribute to the characteristic bitter flavor of plants, results indicated that anthocyanin did not adversely affect cattle palatability [3]. This agreed with the results obtained by Hosoda [35], who reported that feeding with the purple pigment from anthocyanin-rich corn result in the same DM intake as the rice straw diet. The OM, CP, and ADF intake were different between rice straw and purple field corn stover and were the greatest in the purple field corn stover group. This could be because purple field corn stover contains higher nutritive value than rice straw. The high CP intake could provide N available for rumen microbes to synthesize their cells with their energy source [36].

Nutrient digestibility of DM, OM, CP, NDF, and ADF are shown in Table 2. It was found that all nutrient digestibility was significant among treatments (*p* < 0.01). The increase in CP might have an effect on the high digestibility of purple field corn stover [3]. It could also be due to the high growth of mycelia in *P. ostreatus* and *V. volvacea*, which increased the protein concentration [37]. Purple field corn stover contains higher anthocyanin concentration, at 2022.1 mg CGE/100g DW, than normal corn varieties [2], which may support rumen bacterial activity to break down feed. High feed digestibility is associated with great bacterial activity. Low rumen protozoa (Table 3) when animals received anthocyanin from purple field corn stover fermented with *P. ostreatus* and *V. volvacea* led to increased cellulolytic bacteria, which appeared to increase feed digestion of DM and OM [38]. Moreover, the current study showed that NDF and ADF digestion was higher in purple field corn stover than in rice straw, but there were no significant differences between *P. ostreatus* and *V. volvacea*. This could be due to reduced lignin indigestible cell wall contents and enhancement of the digestibility of roughage has been elucidated [10]. This could be due the potential of these fungi to reduce indigestible cell wall contents and improve cell wall digestion of rice straw by secondary metabolites such as manganese peroxidase, lignin peroxidase, and laccase enzyme shown in the structural polymer found in rice straw and purple field corn residue [11]. Furthermore, the difference between NDF and ADF might have been underestimated due to a portion of monomers and oligomers of hemicellulose liberated by the fungi being washed out during the NDF analysis [39].

### 3.3. Ruminal Characteristics, Microbial Count and Blood Metabolite

The rumen pH, temperature determined via rumen fluid, and average values ranged from 6.75 to 7.04 and 38.49 to 38.91 °C (Table 3), which is optimal for microbial digestion in rumen [40]. However, the ruminal pH was slightly dropped in hour 4 post feeding. This result indicates the possibility that the microbial digestion of feed could decrease pH value in the rumen. Cherdthong et al. [41] indicated that the ranges of rumen pH were decreased at 4 h after morning feeding.

There was no interaction effect on NH_3_-N at 0 h after feeding (*p* > 0.05). However, there were significant effect of roughage sources on NH_3_-N at 4 h after feeding. This may be due to the high CP intake when the purple field corn stover was fed, leading to a high amount of CP available from corn for microbial breakdown to NH_3_-N in the rumen. The ruminal NH_3_-N is the main product of protein digestion, which would be utilized by rumen microorganisms in the rumen. Thus, the additional levels of CP in the feed can provide rumen NH_3_-N availability. Hristov et al. [42] demonstrated that protein could degrade into NH_3_-N by microbial extracellular enzymes and deamination. Likewise, a previous report indicated that NH_3_-N was increased by roughage fermented with white rot as compared with a control in vitro [12]. Moreover, the high NH_3_-N concentration in the roughage fermented with *P. ostreatus* was probably influenced by the high CP level, which is a supplemental source of N. In addition, *P. ostreatus* and *V. volvacea* are composed of CP, and there is a probability of additional protein supply to the cattle.

Bacterial population was changed by feeding with purple field corn stover fermented with *P. ostreatus* and *V. volvacea*. On the other hand, the number of protozoa was reduced by approximately 33% at 4 h after feeding with purple field corn stover (*p* < 0.01). There are at least three reasons for this: (i) the fungi could be promoted by the synergy that occurs with rumen microbes. It can penetrate the plant tissue as a result of their filamentous growth, have a broad range of highly active enzymes, and is the only known rumen microorganism with exo-acting cellulase activity. Furthermore, (ii) the anthocyanin contained in purple field corn stover may inhibit the protozoa activity, likely by binding to the protozoal membranes as reported by Cieślak [43]. Moreover, (iii) gallic acid in anthocyanin can bind to the glutamate-gated chloride channels in the nervous system of *Caenorhabditis elegans* and initiate the hyperpolarization of the cell membranes and excitation of muscles. These events finally result in protozoa paralysis and death [44]. Thus, lowering protozoa could result in enhancing the bacteria population in the rumen [45].

The BUN did not change because of the different diets (*p* > 0.05) (Table 3). The means of BUN concentration in cattle fed rice straw and purple field corn stover fermented with *P. ostreatus* and *V. volvacea* were in the comfortable range (12–13 mg/dl), as demonstrated by Cherdthong and Wanapat [41].

### 3.4. Volatile Fatty Acid Profile and CH_4_ Production

Feeding rice straw and purple field corn stover fermented with *P. ostreatus* and *V. volvacea* had no influence on total VFA and butyrate concentration (*p* > 0.05). The acetic acid concentration at 4 h after morning feeding was higher in cattle fed purple field corn stover fermented with fungi. This is likely due to higher NDF and ADF digestion of purple field corn stover. Also, Pino et al. [46] found that the low feed quality diet containing grass hay had higher acetic acid proportions compared with high feed quality diets. Nevertheless, propionic acid concentration was different between roughage sources (*p* < 0.01) enhanced with purple field corn stover fermented with *P. ostreatus* and *V. volvacea*. Ruminal bacteria ferment carbohydrates and break down into simple sugars. The microbes use these sugars as an energy source for their own growth and produce propionic acid [47,48]. Moreover, purple field corn stover had high anthocyanin and gallic acid, which could inhibit CH_4_ synthesis by changing hydrogen from the CH_4_ pathway to propionic acid form [5,36]. An alternative electron sink for a metabolic route to dispose the reduced power has to happen. Furthermore, Newbold et al. [45] proposed that the succinate-propionate pathway could possibly lead to propionic acid production.

In addition, CH_4_ production decreased by 15% with purple field corn stover fermented with *P. ostreatus* and *V. volvacea* compared to rice straw (Table 3). This was probably because lovastatin might inhibit the HMG-CoA reductase enzyme in methanogenic bacteria by interfering with the assembly of isoprenoid chains needed for membrane phospholipid synthesis [49,50]. Moreover, lovastatin might inhibit the expression of the F420-dependent NADH reductase gene. The gene produces the enzyme for the transfer of the methyl group from methyl- H4MPT to HS-COM [51]. Methyl coenzyme-M reductase (mcr) is the key enzyme in the methanogenesis process [5,41]. These results were supported by our previous in vitro test, which showed that fermenting roughage with *P. ostreatus* and *V. volvacea* significantly reduced CH_4_ production by approximately 10.83% to 11.32% compared to the untreated substrates [12]. Likewise, Jahromi [14] reported that lovastatin in fermented rice straw is effective at suppressing 96.43% of the CH_4_ emission. In addition, protozoal rumen provides the perfect host for methanogens to protect their activities (i.e., low O_2_ pressure and high H_2_ availability) [52] and to protect them from oxygen toxicity [53]. This relations is a typical case of interspecies H_2_ transfer that favors the methanogenic bacteria and the protozoal populations [54].

### 3.5. Nitrogen Balances and Microbial Protein Synthesis

There were significant differences in all nitrogen balances parameters (*p* < 0.05), except the fecal N excretion (*p* > 0.05) which were not changed (Table 4). N intake, N excretion, and N absorption of purple field corn stover fermented with *P. ostreatus* and *V. volvacea* were greater (*p* < 0.01) than the fermented rice straw. However, the allantoin excretion was in general lower compared to those found by other authors [55,56], even if it is to underline that it is higher in animals fed PPCS compared to rice straw, according to the higher DM and OM digestibility [39]. Moreover, allantoin excretion and absorption were higher with the purple field corn stover fermented with *P. ostreatus* and *V. volvacea*. Furthermore, MCP and EMNS were enhanced when purple field corn stover fermented with *P. ostreatus* and *V. volvacea* was fed compared to rice straw group. This might be because the high CP and carbohydrates in purple field corn stover have the potential to improve the ruminal fermentation toward maximizing microbial protein production [57]. Furthermore, anthocyanin could support a synchronized release of nitrogen and carbohydrates from purple field corn stover, which is responsible for microbial efficiency enhancement [36]. More reasons could be that protozoa engulf other microbes as their main nutrient source [58] and as a result, defaunation could be more rumen efficient in terms of proteosynthesis enhancing the duodenal flow of microbe mass. Similarly, defaunation also enhanced the efficiency of efficiency microbial protein synthesis (EMPS) (+6.8%) as a result of both, and a greater microbial synthesis was reported [59].

## 4. Conclusions

Based on this study, it could be summarized that *P. ostreatus* or *V. volvacea* can enhance the quality of purple field corn stover, modulate rumen fermentation, feed digestion, and efficiency of microbial N synthesis as well as reduce methane production in Thai native beef cattle. However, these findings should be further elucidated regarding milk yield to determine the influence of diet on dairy cow production.

## Figures and Tables

**Table 1 animals-09-01084-t001:** Ingredient and chemical composition and plant secondary metabolites of concentrate, rice straw (RS) and purple field corn stover (PPCS) used in the experiment.

Parameters	Concentrate	Before fermented		After fermented			
		RS	PPCS	RS		PPCS	
				*P. ostreatus*	*V. volvacea*	*P. ostreatus*	*V. volvacea*
Ingredients, (%DM)							
Cassava chip	55.00						
Rice bran	11.00						
Palm kernel meal, solvent	13.50						
Coconut kernel meal, solvent	12.90						
Molasses, liquid	2.00						
Urea	2.60						
Pure sulfur	1.00						
Mineral premix *	1.00						
Salt	1.00						
Chemical composition							
Dry matter, %	87.40	90.10	94.40	55.80	56.23	55.65	56.14
Organic matter, %DM	94.70	91.80	96.50	91.80	91.50	95.39	94.74
Neutral detergent fiber, %DM	12.20	65.40	70.60	61.97	61.32	67.60	64.45
Acid detergent fiber, %DM	8.40	40.20	53.20	37.57	37.22	42.37	43.87
Crude protein, %DM	14.20	2.60	4.90	6.95	7.38	9.47	9.56
Plant secondary metabolites							
Anthocyanin, %DM	-	-	11.8	0.53	0.57	6.25	5.76
Lovastatin, g/kg	-	-	-	33.92	33.96	32.49	34.90

* Minerals and vitamins (each kg contains): Vitamin A: 10,000,000 IU; Vitamin E: 70,000 IU; Vitamin D: 1,600,000 IU; Fe: 50 g; Zn: 40 g; Mn: 40 g; Co: 0.1 g; Cu: 10 g; Se: 0.1 g; I: 0.5 g. * RS = rice straw; PPCS = purple field corn stover. Anthocyanin content (cyanidin-3-glucoside equivalents, mg/L) = (A × MW × DF × 103)/(ε × 1); where A = (A510 nm–A700 nm)pH1.0–(A510 nm–A700 nm)pH 4.5; MW (molecular weight) = 449.2 g/mol for cyanidin-3-glucoside (cyd-3-glu); DF = dilution factor; 1 = pathlength in cm., ε = 26,900 molar extinction coefficient, in L·mol−1·cm−1, for cyd-3-glu and 103 = factor for conversion from g to mg.

**Table 2 animals-09-01084-t002:** Feed intake and digestibility of cattle feeding rice straw (RS) and purple field corn stove (PPCS) fermented with *Pleurotus ostreatus* and *Volvariella volvacea*.

Parameters	RS		PPCS		SEM	Contrast		
	*P. ostreatus*	*V. volvacea*	*P. ostreatus*	*V. volvacea*		R	WRF	R × WRF
DM intake								
Roughage								
kg/d	3.54	3.39	3.23	3.80	0.18	0.80	0.30	0.08
g/kg BW^0.75^	87.69	83.52	79.27	92.12	5.44	0.98	0.44	0.14
Concentrate								
kg/d	1.39	1.40	1.41	1.43	0.04	0.56	0.79	0.95
g/kg BW^0.75^	34.32	34.38	34.48	34.57	0.27	0.55	0.79	0.95
Total								
kg/d	4.93	4.79	4.65	5.23	0.18	0.69	0.27	0.07
g/kg BW^0.75^	122.01	117.91	113.75	126.69	5.31	0.96	0.42	0.13
Nutrients intake, kg/d								
Dry matter	4.93	4.79	4.65	5.23	0.18	0.69	0.27	0.07
Organic matter	4.55 ^a^	4.21 ^a^	4.73 ^b^	4.90 ^b^	0.17	0.02	0.64	0.17
Crude protein	0.39 ^a^	0.38 ^a^	0.49 ^b^	0.51 ^b^	0.02	<0.01	0.75	0.52
Neutral detergent fiber	2.91	2.65	3.14	3.13	0.20	0.11	0.53	0.56
Acid detergent fiber	1.82 ^a^	1.66 ^a^	2.01 ^b^	2.13 ^b^	0.14	0.03	0.91	0.33
Digestibility coefficients, %								
Dry matter	60.54 ^a^	60.20 ^a^	64.85 ^b^	65.16 ^b^	0.36	<0.01	0.96	0.39
Organic matter	64.96 ^a^	63.43 ^a^	67.93 ^b^	67.78 ^b^	0.99	<0.01	0.41	0.50
Crude protein	62.55 ^a^	61.23 ^a^	65.21 ^b^	65.36 ^b^	0.88	<0.01	0.52	0.42
Neutral detergent fiber	41.43 ^a^	42.42 ^a^	47.81 ^b^	46.56 ^b^	0.80	<0.01	0.73	0.25
Acid detergent fiber	37.24 ^a^	35.94 ^a^	41.73 ^b^	41.72 ^b^	0.91	<0.01	0.74	0.29

* RS = rice straw; PPCS = purple field corn stover; R = roughage; WRF = white rot fungi; R × WRF = interaction between roughage and white rot fungi; SEM, standard error of the mean. ^a,b^ Means differing letters across rows indicate significant differences (*p* < 0.05).

**Table 3 animals-09-01084-t003:** Ruminal ecology, ammonia nitrogen, microorganism, volatile fatty acid profile, and methane estimation of cattle feeding rice straw and purple field corn stove treated *Pleurotus ostreatus* and *Volvariella volvacea*.

Parameters	RS		PPCS		SEM	Contrast		
	*P. osteratus*	*V. volvacea*	*P. osteratus*	*V. volvacea*		R	WRF	R × WRF
Rumen ecology								
Ruminal temperature, °C								
0 h post feeding	38.30	38.47	38.55	38.12	0.26	0.85	0.63	0.28
4 h post feeding	39.33	39.36	39.26	38.86	0.28	0.34	0.52	0.46
Mean	38.82	38.91	38.91	38.49	0.24	0.52	0.53	0.32
Ruminal pH								
0 h post feeding	7.08	7.19	7.12	6.97	0.18	0.63	0.91	0.51
4 h post feeding	7.06	7.06	6.95	6.52	0.18	0.11	0.26	0.26
Mean	6.95	7.00	7.04	6.75	0.28	0.58	0.43	0.26
Ammonia nitrogen (NH_3_-N) concentration, mg/dL								
0 h post feeding	12.95	12.60	14.36	14.36	1.51	0.31	0.91	0.91
4 h post feeding	14.71 ^a^	14.51 ^a^	16.75 ^b^	17.60 ^b^	0.28	<0.01	0.27	0.08
Mean	14.53	14.43	14.85	15.11	0.80	0.54	0.92	0.83
Volatile fatty acid profile, mol/100 mol								
Acetic acid								
0 h post feeding	53.52	53.51	55.14	56.63	2.70	0.39	0.78	0.78
4 h post feeding	55.44 ^a^	55.17 ^a^	58.70 ^b^	62.19 ^b^	2.14	0.03	0.46	0.39
Mean	54.48	54.59	56.85	54.59	2.04	0.10	0.57	0.57
Propionic acid								
0 h post feeding	28.51 ^a^	28.99 ^a^	33.66 ^b^	32.09 ^b^	1.51	0.03	0.72	0.51
4 h post feeding	32.59 ^a^	32.26 ^a^	36.37 ^b^	35.47 ^b^	1.48	0.01	0.68	0.85
Mean	31.12 ^a^	30.62 ^a^	35.01 ^b^	33.78 ^b^	1.29	0.01	0.51	0.78
Butyric acid								
0 h post feeding	12.26	11.10	10.14	11.00	2.00	0.59	0.94	0.62
4 h post feeding	12.78	12.12	10.88	12.72	2.20	0.77	0.79	0.58
Mean	12.52	11.06	10.51	10.72	1.52	0.45	0.69	0.59
Total VFA								
0 h post feeding	98.38	96.54	101.66	103.02	3.16	0.14	0.94	0.62
4 h post feeding	96.73 ^a^	96.29 ^a^	103.99 ^b^	107.00 ^b^	1.98	<0.01	0.53	0.40
Mean	98.57 ^a^	95.69 ^a^	101.60 ^b^	102.90 ^b^	1.81	0.01	0.66	0.27
Blood urea nitrogen, mg/dL								
0 h post feeding	16.75	15.00	13.25	14.25	1.63	0.21	0.82	0.41
4 h post feeding	18.25	14.00	16.25	16.50	1.44	0.86	0.19	0.14
Mean	17.50	14.50	14.75	15.37	1.18	0.44	0.33	0.15
Methane estimation *, mM/L								
0 h post feeding	20.75 ^a^	21.05 ^a^	18.12 ^b^	18.72 ^b^	1.09	0.04	0.68	0.89
4 h post feeding	23.68 ^a^	25.76 ^a^	20.04 ^b^	19.44 ^b^	1.17	<0.01	0.54	0.27
Mean	22.21 ^a^	23.40 ^a^	19.08 ^b^	18.87 ^b^	1.09	<0.01	0.66	0.53
Microbial populations, cell/mL								
Bacteria, ×10^10^								
0 h post feeding	4.27 ^a^	5.00 ^a^	8.25 ^b^	6.53 ^b^	1.11	0.04	0.67	0.31
4 h post feeding	4.87 ^a^	5.15 ^a^	9.65 ^b^	8.84 ^b^	2.02	0.05	0.89	0.79
Mean	5.51 ^a^	5.07 ^a^	8.65 ^b^	7.82 ^b^	1.06	0.01	0.55	0.85
Protozoa, ×10^6^								
0 h post feeding	7.20	5.50	3.00	4.83	1.58	0.18	0.91	0.32
4 h post feeding	6.62 ^a^	4.50 ^a^	3.50 ^b^	3.96 ^b^	0.66	0.01	0.25	0.09
Mean	6.91 ^a^	5.00 ^a^	3.25 ^b^	3.73 ^b^	0.78	0.02	0.34	0.16

* RS = rice straw; PPCS = purple field corn stover; R = roughage; WRF = white rot fungi; R × WRF = interaction between roughage and white rot fungi; CH_4_ = (0.45 × acetic acid) − (0.275 × propionic acid) + (0.40 × butyric acid) (Moss et al., 2000). SEM, standard error of the mean. ^a,b^ Means differing letters across rows indicate significant differences (*p* < 0.05).

**Table 4 animals-09-01084-t004:** Nitrogen (N) balance and purine derivative of cattle feeding rice straw and purple field corn stove treated *Pleurotus ostreatus* and *Volvariella volvacea*.

Parameters	Rice Straw		Purple corn stove		SEM	Contrast		
	*P. osteratus*	*V. volvacea*	*P. osteratus*	*V. volvacea*		R	WRF	R × WRF
N intake (NI), g/d	62.91 ^a^	62.77 ^a^	77.93 ^b^	81.90 ^b^	3.84	<0.01	0.62	0.60
Total N excretion, g/d	50.23 ^a^	51.02 ^a^	55.56 ^b^	58.23 ^b^	0.58	0.02	0.88	0.65
Fecal N excretion, g/d	15.07	15.31	16.67	17.47	1.89	0.18	0.68	0.44
Urinary N excretion, g/d	35.16 ^a^	35.71 ^a^	38.89 ^b^	40.76 ^b^	1.35	0.01	0.58	0.71
N absorption, g/d	47.84 ^a^	47.46 ^a^	61.26 ^b^	64.43 ^b^	2.36	<0.01	0.83	0.36
% of N absorption	30.10 ^a^	29.79 ^a^	47.74 ^b^	52.77 ^b^	1.55	0.02	0.65	0.89
N retention, g/d	12.68 ^a^	11.75 ^a^	22.37 ^b^	23.67 ^b^	1.23	0.03	0.33	0.47
% of N retention to N intake	20.16 ^a^	18.72 ^a^	28.71 ^b^	28.90 ^b^	2.19	0.05	0.53	0.77
PD, mmol/d								
Allantoin excretion	80.55	80.27	88.17	88.15	0.51	<0.01	0.81	0.84
Allantoin absorption	74.62 ^a^	74.27 ^a^	82.92 ^b^	82.77 ^b^	0.58	<0.01	0.67	0.86
Microbial crude protein, g/d	275.11 ^a^	274.16 ^a^	301.14 ^b^	301.07 ^b^	3.97	<0.01	0.73	0.19
EMPS *	64.16 ^a^	64.50 ^a^	68.20 ^b^	68.34 ^b^	0.92	0.03	0.34	0.06

* RS = rice straw; PPCS = purple field corn stover; R = roughage; WRF = white rot fungi; R × WRF = interaction between roughage and white rot fungi; EMPS: efficiency of microbial protein synthesis. N: Nitrogen. Microbial crude protein (g/d) = 3.99 × 0.856 × mmoles of purine derivatives excreted. Efficiency of microbial N synthesis (EMNS, g/kg of organic matter (OM) digested in the rumen (OMDR) = ((MCP (g/d) × 1000)/DOMR (g)), assuming that rumen digestion was 650 g/kg OM of digestion in total tract. SEM, standard error of the mean. ^a,b^ Means differing letters across rows indicate significant differences (*p* < 0.05).

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
