# Peer review of "Pleurotus Ostreatus and Volvariella Volvacea Can Enhance the Quality of Purple Field Corn Stover and Modulate Ruminal Fermentation and Feed Utilization in Tropical Beef Cattle"

_animals, 2019, doi:10.3390/ani9121084_

Round 1
Reviewer 1 Report
Please find attached file

Author Response
Dear Editor-in-Chief Animals journal,
Manuscript ID: animals-645503
Title: Pleurotus ostreatus and Volvariella volvacea can enhance the quality of purple field corn stover and modulate ruminal fermentation and feed utilization in tropical beef cattle
We highly appreciated all the comments and suggestions made by the two reviewers. Above all, the authors felt that all points made were very useful and have incorporated most of the corrections where necessary as suggested in order to make the manuscript ready for possible publication. All those corrected and modified appear in track change mode in the manuscript. Please see information given by the authors following the suggestions and comments made by the three reviewers.
With the above information we would like to resubmit our paper for your kind considerations for a possible publication in Animals journal.
We again wish to thank you very much for your kind attention and support.
Sincerely yours,
Assoc. Prof. Dr. Anusorn Cherdthong
Corresponding author
Contact: [email protected]
Response to the Reviewer 1
The aim of manuscript was to evaluate the role of two species of white-rot fungal mushroom on improving forage utilization, rumen ecology and methane emission in beef cattle. This is particularly important in tropical areas where animal sare usually fed with low-quality forage.
The paper is well written with clear discussion o results. It needs littel modifications before the publication:
Introduction : the authors have to add informations on the Lovastatin
Response: We have already provided as “Moreover, lovastatin can be produced by various white-rot fungi by controlling the culture conditions. The lovastatin has been demonstrated to effectively inhibit the HMG-CoA reductase enzyme in methanogenic bacteria by interfering with the assembly of isoprenoid chains needed for membrane phospholipid synthesis and thus selectively depressing growth of methanogenic bacteria without affecting other ruminal microorganism community [12-14].” Please see in page 2 of the manuscript.
Results and discussion: to improve the discussion on N balance I suggets to report that the allantoin excretion was in general lower compared to those found by other authors (Cutrignelli et al., 2007; Jardstedr at al., 2017), even if it is to underline that it is higher in animals fed PPCS compared to rice straw, according to the higher DM and OM digestibility (Sudekum et al., 2006).
Response: We have already provided as “However, the allantoin excretion was in general lower compared to those found by other authors (Cutrignelli et al., 2007; Jardstedr et al., 2017), even if it is to underline that it is higher in animals fed PPCS compared to rice straw, according to the higher DM and OM digestibility (Sudekum et al., 2006).” Please see in page 12 in the manuscript.
References: pleas add the following Cutrignelli M.I., Piccolo G., D’Urso S., Calabrò S., Bovera F., Tudisco R., Infascelli F., 2007. Urinary excreti on of purine derivatives in dry buffalo and Fresian cows. Italian Journal of Animal Science, 6, suppl 2., 563-566
Response: We have already added and please see in the Reference section of the manuscript.
Jardstedt M., Hessle A., Nørgaard P., Richardt W., Nadeau E. 2017. Feed intake and urinary excretion of nitrogen and purine derivatives in pregnant suckler cows fed alternative roughage-based diets. Livestock Science, 202, 82-88
Response: We have already added and please see in the Reference section of the manuscript.
Südekum K.H., Brusemeister F., Schroder A., Stangassinger M. 2006. Effects of amount of intake and stage of forage maturity on urinary allantoin excretion and estimated microbial crude protein synthesis in the rumen of steers. J. Anim. Physiol. Anim. Nutr., 90, 136-145
Response: We have already added and please see in the Reference section of the manuscript.

Reviewer 2 Report
Animals-645503: “Pleurotus ostreatus and Volvariella volvacea can enhance the quality of purple field corn stover and modulate ruminal fermentation and feed utilization in tropical beef cattle”. The topic of the MS is very interesting, and it can bring value to the scientific community. I commend authors presenting a fairly novel study; However, in this reviewer's opinion, there are some issues that need to be addressed before we move forward. They are listed below:
Author hypothesized that: “…feeding purple field corn stover fermented with P. ostreatus or V. volvacea will increase feed utilization and reduce CH4 production in beef cattle…” However, this hypothesis in its present form seem to be missing something. For example, how authors will compare that both compounds will increase feed utilization and CH4 production in beef cattle if both roughages source received both compounds and there is no control treatment to compared the effect of these compounds?
In table 1 author present stats analysis for diet ingredients. However, in my opinion there is no sample replication that would allow authors to perform stats for the diets. I would suggest author only reporting the chemical composition w/o stats.
There is no description of how urine samples were collected. Please, describe? Why author did not evaluate uric acid concentration in urine to estimate the microbial protein synthesis??
Table 2. and all other tables. Please describe the acronyms of RS and PPCS for all tables. In addition, describe in the footnote the contrasts.
Table 4. Nitrogen retention values and other values in this tables does not looks right. For example, what would explain more than 60% of nitrogen retention (% of the diet) with a low-quality diet? Usually, for efficiency ruminants’ animals these values in literature are around 30%. Another example, if the animals consumed on average 446 g/d of CP. How did the bacteria in the rumen synthesized on average 424.6 g/d of CP. This value is about 95% of the total CP provided from the diet. Did you estimate the RDP and RUP of your diets?? I would suggest author revising all these values again. In addition, author needs to explain why the only measure allantoin to estimated the total PD excretion and absorption.
In the conclusion authors stated that “…feeding approach regime to cattle and protecting environment pollution in agricultural industry…” In this reviewers’ s opinion this statement is too strong and is out of the scope of the present study. Authors did not evaluate environment pollution impact in the present study. I suggest authors focusing their conclusion based on the variables that were measured in the present study.
Minor comments:
Line 23: “…beef cattle (100 + 30 kg of body…” Would not be “…(100 ± 30 kg of body…” ??
Line 50. Change “…synthesis…” to “…synthesized…”
Line 54. Are you sure that this statement is correct?? “…with 90% from cattle and cows as an effect of feed digestion…”
Lines 64-65. You stated that “…many studies have presented the various biological treatment….” however, you only cited one study. Please, add at least 2 more studies.
Line 67. I believe that the references [10] refers to reference [11]. Please, double check.
Pg 5. Line 25. Which procedure did use to estimate fecal output?? Total collection or marker? Please describe.
Pg5. Line 40. Cite the reference for NH3-N concentration determination.
Pg6 Lines 60-67. Author spent a section describing the methods used to determine Lovastatin content. However, there is no mention in the intro section why is important to measure this compound and to understand the rationale of measuring it.
Author Response
Dear Editor-in-Chief Animals journal,
Manuscript ID: animals-645503
Title: Pleurotus ostreatus and Volvariella volvacea can enhance the quality of purple field corn stover and modulate ruminal fermentation and feed utilization in tropical beef cattle
We highly appreciated all the comments and suggestions made by the two reviewers. Above all, the authors felt that all points made were very useful and have incorporated most of the corrections where necessary as suggested in order to make the manuscript ready for possible publication. All those corrected and modified appear in track change mode in the manuscript. Please see information given by the authors following the suggestions and comments made by the three reviewers.
With the above information we would like to resubmit our paper for your kind considerations for a possible publication in Animals journal.
We again wish to thank you very much for your kind attention and support.
Sincerely yours,
Assoc. Prof. Dr. Anusorn Cherdthong
Corresponding author
Contact: [email protected]
Response to the Reviewer 2
Animals-645503: “Pleurotus ostreatus and Volvariella volvacea can enhance the quality of purple field corn stover and modulate ruminal fermentation and feed utilization in tropical beef cattle”. The topic of the MS is very interesting, and it can bring value to the scientific community. I commend authors presenting a fairly novel study; However, in this reviewer's opinion, there are some issues that need to be addressed before we move forward. They are listed below:
Author hypothesized that: “…feeding purple field corn stover fermented with P. ostreatus or V. volvacea will increase feed utilization and reduce CH4 production in beef cattle…” However, this hypothesis in its present form seem to be missing something. For example, how authors will compare that both compounds will increase feed utilization and CH4 production in beef cattle if both roughages source received both compounds and there is no control treatment to compared the effect of these compounds?
Response: We have revised as “The hypothesis is that feeding cattle with purple field corn stover fermented with P. ostreatus or V. volvacea will increase feed utilization and reduce CH4 production when compared to rice straw.” Please see in page 2 of the manuscript.
In table 1 author present stats analysis for diet ingredients. However, in my opinion there is no sample replication that would allow authors to perform stats for the diets. I would suggest author only reporting the chemical composition w/o stats.
Response: Thanks for your suggestion and now we have removed the present stats analysis for diet ingredients. Please see in Table 1.
There is no description of how urine samples were collected. Please, describe? Why author did not evaluate uric acid concentration in urine to estimate the microbial protein synthesis??
Response: We have provided detail as “The animals were in an adjusting dietary phase for the first 14 days, and the last 7 days of each phase were for feces and urine sampling as cattle were transferred to metabolic stalls for determination of feed digestion and nitrogen balance by using for total collection method. Urine was sampled in 10-L vessels containing 10% H2SO4, to ensure that the pH was decreased below 3.0 in order to prevent bacterial destruction of purine derivative and creatinine. Urine vessels were replaced every 24 h, the volume of acidified urine was measured and a subsample of 100 mL per animal and was immediately frozen at −20 °C.” Please see in page 5 of manuscript. Furthermore, present study did not determine for uric acid concentration due to its do not including in equation of Chen et al. as follow; the concentration of microbial purine absorbed (X mmol/day) corresponding to the purine derivatives excreted (Y mmol/day) was estimated based on the relationship derived by Chen and Gomes [17]: Y = 0.85X+ (0.385W0.75),where Y is the excretion of purine derivatives (mmol/day), X the microbial purines absorbed (mmol/day) and W is body weight of animal (g/kg BW0.75). Thus, uric acid concentration may not need evaluate and spend more budget analysis.
Table 2. and all other tables. Please describe the acronyms of RS and PPCS for all tables. In addition, describe in the footnote the contrasts.
Response: Thanks so much and we have already described regarding the comments. Please see in all Tables.
Table 4. Nitrogen retention values and other values in this tables does not looks right. For example, what would explain more than 60% of nitrogen retention (% of the diet) with a low-quality diet? Usually, for efficiency ruminants’ animals these values in literature are around 30%. Another example, if the animals consumed on average 446 g/d of CP. How did the bacteria in the rumen synthesized on average 424.6 g/d of CP. This value is about 95% of the total CP provided from the diet. Did you estimate the RDP and RUP of your diets?? I would suggest author revising all these values again. In addition, author needs to explain why the only measure allantoin to estimated the total PD excretion and absorption.
Response: Thank you for the great comments. Now, we have rechecked and recalculated regarding to your comments already. The data modification is presented in Table 4 and please see in manuscript.
In the conclusion authors stated that “…feeding approach regime to cattle and protecting environment pollution in agricultural industry…” In this reviewers’ s opinion this statement is too strong and is out of the scope of the present study. Authors did not evaluate environment pollution impact in the present study. I suggest authors focusing their conclusion based on the variables that were measured in the present study.
Response: We have revised as “Base on this study, it could be summarized that P. ostreatus or V. volvacea can enhance the quality of purple field corn stover, modulate rumen fermentation, feed digestion and efficiency of microbial N synthesis as well as reduce methane production in Thai native beef cattle. However, these findings should be further elucidated regarding milk yield to determine the influence of diet on dairy cow production.” Please see in page 14 of manuscript.
Minor comments:
Line 23: “…beef cattle (100 + 30 kg of body…” Would not be “…(100 ± 30 kg of body…” ??
Response: We have already changed.
Line 50. Change “…synthesis…” to “…synthesized…”
Response: We have already changed.
Line 54. Are you sure that this statement is correct?? “…with 90% from cattle and cows as an effect of feed digestion…”
Response: We have removed this sentient out from manuscript. Please see in page 2.
Lines 64-65. You stated that “…many studies have presented the various biological treatment….” however, you only cited one study. Please, add at least 2 more studies.
Response: We have modified and added 3 research works and please see in page 2.
Line 67. I believe that the references [10] refers to reference [11]. Please, double check.
Response: We have modified and please see in page 2.
Pg 5. Line 25. Which procedure did use to estimate fecal output?? Total collection or marker? Please describe.
Response: We have modified as “The study was separated into 4 periods, which each consisted of 21 days. The animals were in an adjusting dietary phase for the first 14 days, and the last 7 days of each phase were for feces and urine sampling as cattle were transferred to metabolic stalls for determination of feed digestion and nitrogen balance by using for total collection method. The fecal samples were sampled about 5% of total fresh weight and divided into two parts; first part for DM analysis every day and second part was kept in refrigerator. Urine was sampled in 10-L vessels containing 10% H2SO4, to ensure that the pH was decreased below 3.0 in order to prevent bacterial destruction of purine derivative and creatinine. Urine vessels were replaced every 24 h, the volume of acidified urine was measured and a subsample of 100 mL per animal and was immediately frozen at −20 °C. Feces and urine were pooled by cattle at the end of each period for chemical analysis. ” Please see in Manuscript page 5.
Pg5. Line 40. Cite the reference for NH3-N concentration determination.
Response: We have indicated. Please see in Manuscript page 5.
Pg6 Lines 60-67. Author spent a section describing the methods used to determine Lovastatin content. However, there is no mention in the intro section why is important to measure this compound and to understand the rationale of measuring it.
Response: We have already provided as “Moreover, lovastatin can be produced by various white-rot fungi by controlling the culture conditions. The lovastatin has been demonstrated to effectively inhibit the HMG-CoA reductase enzyme in methanogenic bacteria by interfering with the assembly of isoprenoid chains needed for membrane phospholipid synthesis and thus selectively depressing growth of methanogenic bacteria without affecting other ruminal microorganism community [12-14].” Please see in page 2 in the manuscript.
Thank you!

Round 2
Reviewer 2 Report
Thanks for addressing all the comments accordingly. However, I highly recommend authors revisiting the Chen and Gomes article. Total purine derivatives (PD) are the sum of Allantoin + uric acid for cattle, and for sheep PD are the sum of Allantoin + uric acid + hypoxanthine + xanthine.